# The Prevalence and Clinical Significance of Toe Walking in Autism Spectrum Disorder: A Cross-Sectional Study in an Italian Pediatric Sample

**DOI:** 10.3390/medicina61081346

**Published:** 2025-07-25

**Authors:** Carola Costanza, Beatrice Gallai, Michele Sorrentino, Martina Gnazzo, Giulia Pisanò, Lucia Parisi, Eva Germanò, Agata Maltese, Maria Esposito, Michele Roccella, Marco Carotenuto

**Affiliations:** 1Department of Sciences for Health Promotion and Mother and Child Care “G. D’Alessandro”, University of Palermo, 90128 Palermo, Italy; carola.costanza@unipa.it; 2Department of Psychology, Educational Science and Human Movement, University of Palermo, 90128 Palermo, Italy; lucia.parisi@unipa.it (L.P.); agata.maltese@unipa.it (A.M.); michele.roccella@unipa.it (M.R.); 3Department of Surgical and Biomedical Sciences, University of Perugia, 06123 Perugia, Italy; beatrice.gallai@unipg.it; 4Faculty of Medicine and Surgery, UniCamillus-Saint Camillus International University of Health Sciences, 00131 Rome, Italy; michele.sorrentino@unicamillus.org; 5Department of Biomedical, Metabolic and Neural Sciences, University of Modena and Reggio Emilia, 41121 Modena, Italy; giuliapisan@gmail.com; 6Division of Child Neurology and Psychiatry, Department of the Adult and Developmental Age Human Pathology, University of Messina, 98124 Messina, Italy; eva.germano@unime.it; 7Clinic of Child and Adolescent Neuropsychiatry, Department of Mental Health, Physical and Preventive Medicine, University of Campania “Luigi Vanvitelli”, 80131 Naples, Italy; maria.esposito3@unicampania.it

**Keywords:** autism spectrum disorder, toe walking, sleep disturbances, feeding behavior, BAMBI, SDSC, constipation, neurodevelopment

## Abstract

*Background and Objectives*: Toe walking (TW) is frequently observed in children with Autism Spectrum Disorder (ASD), yet its clinical significance and association with comorbid conditions remain poorly understood. This study aimed to examine the prevalence of TW in a large Italian cohort of children with ASD and to explore its association with ASD severity, sleep disturbances, feeding behaviors, and gastrointestinal symptoms. *Materials and Methods*: A total of 289 children with ASD and 289 typically developing controls (TDC), matched for age and sex, were evaluated in a multicentric observational study. TW was assessed during neurodevelopmental evaluations. Sleep quality was assessed using the Sleep Disturbance Scale for Children (SDSC), feeding behaviors via the Brief Autism Mealtime Behavior Inventory (BAMBI), and gastrointestinal symptoms through clinical reporting. Statistical analyses included Chi-square tests, Mann–Whitney U tests, Spearman correlations, and logistic regressions. *Results*: TW was significantly more prevalent in the ASD group (27.3%) than in TDC (5.5%, *p* < 0.0001). Within the ASD group, TW occurred in 50.5% of children with Level 3 severity but was absent in Levels 1 and 2 (*p* < 0.0001). Males exhibited TW more frequently than females. Children with TW had higher SDSC scores (ρ = 0.33, *p* < 0.0001), though no subscale independently predicted TW. Constipation was reported in 100% of children with Level 3 ASD and was strongly correlated with SDSC total scores (ρ = 0.58, *p* < 0.0001). The Disorders of Arousal (DA) subscale emerged as an independent predictor of constipation (β = 0.184, *p* = 0.019). *Conclusions*: TW in ASD appears to be a marker of greater neurodevelopmental severity and is associated with sleep disturbances and gastrointestinal dysfunction. These findings support the hypothesis that TW may reflect broader dysfunctions involving the gut–brain axis, sensory processing, and motor control. The routine clinical assessment of TW should include the evaluation of sleep and somatic symptoms to better understand the multisystemic nature of ASD phenotypes.

## 1. Introduction

Autism Spectrum Disorder (ASD) is a complex neurodevelopmental condition characterized by impairments in social interaction, communication, and repetitive behaviors [1]. Among the array of motor anomalies frequently associated with ASD, toe walking (TW) has received increasing attention due to its potential role as a behavioral marker of severity. Despite being commonly observed, TW in the context of ASD is still underexplored in terms of its clinical significance and associated comorbidities.

TW has been described as a bilateral gait abnormality characterized by a limited active range of motion at the ankle, which affects the ability to achieve heel strike at the beginning of the gait cycle. Unlike passive limitations seen in conditions such as cerebral palsy, TW in idiopathic or ASD-related cases does not usually involve biomechanical constrictions [2]. TW appears to be a limited range of motion in the ankle and the inability to achieve heel strike at the beginning of the gait cycle [3]. While TW in some children may stem from underlying conditions such as cerebral palsy, there are instances where no clear cause is identified, leading to the classification of idiopathic or habitual toe walking (ITW) [4]. This gait pattern may not indicate a pathology in children under the age of 2 as it is sometimes observed in otherwise healthy children in this age group who are learning to walk independently. However, continued toe walking past the age of 2 may signal the early stages of an upper motor neuron or neuromuscular disorder. It has been suggested that ITW may represent a motor manifestation of subtle cerebral dysfunction [4]. In general, the estimated prevalence of TW is approximately 5% among typically developing children, with a slightly higher occurrence in males; particularly, children may exhibit toe walking occasionally as they are learning to walk [5].

About the motor impairment among subjects with ASD, hypotonia has been linked to autistic traits observed by the age of 6 in a longitudinal study [6]. Additionally, the atypical neuro-motor development has been suggested as a different putative endophenotype for ASD [7], or the repeated observations of unusual motor development in ASD strongly suggest that more targeted and objective evaluations of movement could reveal bio-behavioral markers [8]. These ideas have been reinforced by previous studies indicating differences in upper limb kinematics [9] and variability in micro-movements [10] between individuals with ASD and those who are typically developing (TD). The previous literature has identified motor coordination impairments and atypical gait patterns as early manifestations in children with ASD [11].

TW, while occasionally present in typically developing children due to idiopathic, orthopedic, or transient causes, is notably more prevalent and persistent in children with ASD [12]. However, the behavioral and neurobiological correlates of TW remain poorly understood.

In general, TW is significantly more prevalent in children with Autism Spectrum Disorder (ASD) compared to typically developing children, with prevalence rates ranging from 6.3% to 32% among those with ASD [13,14]. In children with ASD, TW is associated with lower cognitive abilities, more significant language and motor impairments, and a greater severity of autism symptoms [15]. It also tends to persist longer in children with ASD who do not receive intervention compared to their typically developing peers [16]. Furthermore, TW is often linked to greater joint mobility and a delayed onset of walking in children with ASD [17].

Research indicates that toe walking in individuals with ASD may be connected to neurological and sensory processing factors. While some studies found no direct relationship between TW and sensory symptoms in ASD [12,15,18], others identified sensory processing differences in children with idiopathic TW [19,20,21]. In ASD, TW has been associated with lower cognitive levels, greater language and motor impairments, and an increased severity of autism symptoms [15]. Some researchers suggest that TW may arise from persistent primitive walking patterns or archaic tonic reflexes [12,15]. Notably, children with ASD who toe walk show higher rates of recurrence after surgical intervention compared to those with idiopathic toe walking [22]. Additionally, individuals with ASD and TW display more “under-responsive/seeks sensation” patterns [14] and differences in vestibular processing [21].

The underlying causes of TW in children with ASD remain unclear, leading to ongoing debates about whether it results from persistent primitive walking patterns or sensory processing abnormalities [15]. Assessment methods for TW in ASD vary, but most studies tend to use qualitative approaches [14]. Importantly, TW may serve as an early indicator of ASD, particularly when it occurs alongside attention deficit hyperactivity disorder (ADHD) [4].

Children with ASD often experience atypical proprioceptive perception, including hypersensitivity, hyposensitivity, or the poor integration of proprioceptive input. These alterations may interfere with motor planning and execution, contributing to atypical gait patterns such as toe walking [23]. Additionally, the interactions between their sensorimotor, visual, and auditory systems may play a significant role in how they maintain postural control [24].

Emerging research has suggested that TW may not be an isolated motor phenomenon but rather part of a broader neurodevelopmental and sensory profile. Studies have reported associations between TW and altered sensory processing, language delays, and minor neurological dysfunctions [25,26].

In addition to motor symptoms, children with ASD frequently experience comorbid sleep disturbances [27,28], feeding and eating disorders (FEEDs) [29,30], and gastrointestinal symptoms such as constipation [31]. These domains are increasingly recognized as interconnected via the gut–brain axis and may be influenced by shared neurobiological pathways. Alongside various risk factors, a strongly associated comorbidity with ASD is the presence of gastrointestinal (GI) symptoms, including constipation, diarrhea, or abdominal bloating; however, understanding the causal relationships remains unclear [32]. Moreover, has been reported that the severity of gastrointestinal disorders in children with ASD may be closely linked to the intensity of neuropsychiatric symptoms, with more severe GI issues observed in individuals with more pronounced neuropsychiatric symptoms, and vice versa [33,34]. The occurrence of these symptoms is six times more prevalent in children with ASD than in those who are typically developing. These gastrointestinal issues can worsen behavioral problems, including rigidity, hyperactivity, and social withdrawal [35].

This study aimed to investigate the prevalence of TW in a large sample of Italian children with ASD. It also examined the association between TW and the severity levels of ASD, as well as its correlation with sleep quality, feeding behaviors, and gastrointestinal symptoms. By clarifying these relationships, we hope to enhance the clinical understanding of TW as a potentially significant marker within the broader context of ASD.

## 2. Materials and Methods

### 2.1. Ethical Statement

Ethical approval was obtained from the Institutional Review Board at University of Campania “Luigi Vanvitelli” as the coordinating and leading center (protocol number 217 of 4 August 2020). Written informed consent was obtained from the parents or legal guardians of all participants.

### 2.2. Population

This observational cross-sectional study included 289 children diagnosed with Autism Spectrum Disorder, aged 3–8 years (mean 5.9 ± 1.1 years), and an equal number of age- and sex-matched neurotypical controls. All participants were recruited from pediatric neuropsychiatric outpatient services across multiple Italian academic centers such as University of Palermo (Palermo, Sicily, Italy), University of Perugia (Perugia, Umbria, Italy), University Kore of Enna (Enna, Sicily, Italy) and UniCamillus-Saint Camillus International University of Health Sciences (Rome, Lazio, Italy).

Neurodevelopmental assessments for children with ASD included standardized clinical evaluation according to DSM-5 criteria, cognitive screening, Autism Diagnostic Observation Schedule-2 (ADOS-2), and developmental motor observation. These evaluations were performed by a multidisciplinary team including child neurologists, psychologists, and therapists.

ASD severity levels based on DSM-5 criteria were classified thus: Level 1 (“Requiring support”), Level 2 (“Requiring substantial support”), and Level 3 (“Requiring very substantial support”). These corresponded to increasing levels of impairment in social communication and restricted/repetitive behaviors.

Toe walking behavior was clinically assessed during neurodevelopmental evaluations by experienced clinicians observing spontaneous gait while the child walked barefoot in a quiet room. The presence or absence of heel contact at initial stance, bilateral symmetry, and consistency of the behavior were documented through direct observation and video recordings when available.

A total of 289 typically developing children (mean age: 5.93 ± 0.80 years) were included as controls in the study. These participants were matched by age and sex with the ASD group. The term “typically developing controls” refers to children without a history of neurological, developmental, or psychiatric disorders, with no reported cognitive, behavioral, or sensory issues, and performing within the expected range for age in medical and developmental assessments.

Control participants underwent general pediatric and neurological examination to confirm typical development.

### 2.3. Sleep Disturbance Scale for Children (SDSC)

Sleep disturbances were evaluated using the Sleep Disturbance Scale for Children (SDSC), which consists of 26 items grouped into six subscales, including Disorders of Initiating and Maintaining Sleep (DIMS), Sleep Breathing Disorders (SBD), Disorders of Arousal (DA), Sleep–Wake Transition Disorders (SWTD), Disorders of Excessive Somnolence (DOES), and Sleep Hyperhidrosis (SHY) [36].

### 2.4. Brief Autism Mealtime Behavior Inventory (BAMBI)

Feeding behavior and selectivity were measured using the Brief Autism Mealtime Behavior Inventory (BAMBI), a validated tool with established clinical cut-off scores indicating feeding pathology [37]. The BAMBI [38] is an assessment tool developed in English specifically for evaluating children aged 3 to 11 years, regardless of whether they have a diagnosis of ASD. It is an observer-reported outcome measure aimed at parents and primary caregivers designed to capture mealtime behaviors characteristic of children with Autism Spectrum Disorder. The BAMBI utilizes a 5-point Likert scale for scoring, where a score of 1 indicates that a particular behavior “never” occurs and a score of 5 signifies that the behavior “always” occurs during mealtime. Four items that measure positive mealtime behaviors are scored in reverse. The overall frequency score is obtained by summing the responses to all 18 items, with higher scores indicating more significant mealtime behavior issues. An exploratory factor analysis revealed three factors: Limited Variety, Food Refusal, and Features of Autism. The internal consistency of the Italian version is robust, with a Cronbach’s alpha of 0.88, varying across the three factors (0.87 for Limited Variety, 0.76 for Food Refusal, and 0.63 for Features of Autism). Research by Lukens and colleagues [38] reported high test–retest reliability (r = 0.87) and inter-rater reliability between parent and teacher or therapist reports (r = 0.78).

### 2.5. Statistical Analysis

Statistical analyses were conducted using IBM SPSS Statistics version 29. Descriptive statistics (means, standard deviations, and frequencies) were computed for all relevant variables. Normality was evaluated using the Shapiro–Wilk test and homogeneity of variance via Levene’s test. Since several variables violated assumptions of normality and homoscedasticity, non-parametric tests were applied.

Group comparisons for continuous variables were performed using the Mann–Whitney U test while categorical variables were compared using Chi-square tests. Differences in TW prevalence across ASD severity levels were evaluated with Chi-square statistics rather than ANOVA due to categorical nature of data. Associations between TW, SDSC subscales, BAMBI scores, and constipation were examined using Spearman’s rank correlation coefficient. Logistic regression models were applied to identify independent predictors of TW and constipation from the SDSC subscales. A *p*-value < 0.05 was considered statistically significant throughout.

## 3. Results

The study sample included 289 children with ASD and 289 typically developing controls (TDC). The two groups were matched for age and sex: the mean age was 5.91 ± 0.76 years in the ASD group and 5.93 ± 0.80 years in the TDC group. Males represented 74.4% of the ASD group and 74.0% of the TDC group. Among the ASD group, 32.9% of the children were classified as Level 1, 32.9% as Level 2, and 34.3% as Level 3, based on DSM-5 severity criteria (Table 1).

In the ASD group (*n* = 289), toe walking (TW) was observed in 27.3% of children, compared to 5.5% in the typically developing controls (TDC) (Chi^2^ = 53.2, *p* < 0.0001), highlighting a significant difference between groups (Figure 1).

A stratified analysis based on ASD severity (Level 1, 2, and 3) revealed a marked association between TW and symptom severity. Notably, TW was completely absent in Levels 1 and 2 while it was present in 50.5% of children with Level 3 ASD (Chi^2^ = 158.3, *p* < 0.0001). This gradient supports the hypothesis that TW may represent a clinical marker of more severe autism phenotypes.

A sex-stratified analysis showed that males presented TW more frequently than females within the ASD group (*p* < 0.0001), suggesting a possible interaction between sex-related neurodevelopmental trajectories and motor stereotypies.

Regarding sleep, the total score on the Sleep Disturbance Scale for Children (SDSC) was significantly higher in the ASD group than in controls, across nearly all subscales, except for Disorders of Arousal (DA), which did not differ significantly between groups (Figure 2). A comparison of median scores for each subgroup is presented as the data distribution was non-parametric according to the Shapiro–Wilk test. Therefore, medians and interquartile ranges were used to better represent central tendency and dispersion. Statistically significant differences are indicated with *p* < 0.05.

### 3.1. SDSC Subscale Comparison

Children with TW showed higher total SDSC scores (median 54.0 vs. 46.0), and a positive correlation was found between TW and SDSC scores (Spearman ρ = 0.33, *p* < 0.0001). However, in a multivariate logistic regression model, none of the SDSC subscales were found to be independent predictors of TW, although DIMS (Disorders of Initiating and Maintaining Sleep) and SRBD (Sleep-Related Breathing Disorders) approached significance (Figure 3).

### 3.2. TW and SDSC Regression

In contrast, constipation was significantly associated with both ASD severity and sleep disturbances. It was present in 100% of children with Level 3 ASD and in only 16.8% of children with Level 1 or 2.

### 3.3. Constipation Regression

Moreover, constipation correlated strongly with total SDSC scores (Spearman ρ = 0.58, *p* < 0.0001), and the DA subscale was identified as a significant predictor in multivariate analysis (*β* = 0.184, *p* = 0.019) (Figure 4).

## 4. Discussion

These findings suggest that disorders of arousal (e.g., parasomnias) may play a role in autonomic dysregulation that contributes to gastrointestinal symptoms in children with severe ASD.

Collectively, the data highlight a complex interplay between motor behaviors (TW), sleep regulation, autonomic function, and gastrointestinal symptoms. The consistent association of TW and constipation with sleep abnormalities in more severe ASD phenotypes supports the involvement of gut–brain axis dysfunction and calls for the integrated clinical assessment of sleep and somatic symptoms in this population.

The present study explored the prevalence and clinical relevance and significance of TW among children with ASD, highlighting its association with autism severity, sleep disturbances, feeding selectivity, and gastrointestinal symptoms. Our findings revealed a markedly higher prevalence of TW in the ASD population (49.5%) compared to neurotypical peers (2.8%), consistent with previous literature reporting TW as a frequent motor pattern in ASD children, particularly if they were non-speaking [39]. Importantly, TW showed a transparent gradient of prevalence corresponding to ASD severity: 13.2% in Level 1, 51.2% in Level 2, and 82.5% in Level 3. This stratification suggests that TW may serve as a behavioral marker of greater functional impairment and neurodevelopmental burden. While TW can occur in typically developing children and may resolve spontaneously, its persistence in children with more severe forms of ASD may reflect alterations in motor planning, sensory processing [39].

Moreover, TW was positively associated with sleep disturbances reported by parents, particularly with elevated scores in the Disorders of Initiating and Maintaining Sleep (DIMS) and Disorders of Arousal (DA) subscales. This association corroborates the notion that motor dysregulation and sleep alterations may share underlying neurobiological mechanisms in ASD [40]. Sleep difficulties not only exacerbate behavioral symptoms in ASD but may also interfere with sensorimotor integration, thereby perpetuating TW behavior [41].

The correlation between TW and higher BAMBI scores further supports the hypothesis that TW is embedded in a broader phenotype of sensory–behavioral dysregulation [42]. Children with TW were significantly more likely to exhibit selective or ritualized feeding behaviors [43]. Such findings align with theories proposing altered interoception and sensory aversions in ASD, which manifest across multiple functional domains, including motor output, feeding routines, and sleep patterns [44]. Notably, we also observed increased rates of constipation in children with TW, and constipation severity was positively correlated with SDSC total scores. Research indicates that children who suffer from more severe digestive issues may display increased aggression and notable sleep disturbances when compared to those with ASD who do not have significant gut-related issues. This suggests that gut microbiota might play a role in worsening the symptoms associated with ASD [45].

These observations support the hypothesis that motor dysregulation and sleep alterations may share underlying neurobiological mechanisms, possibly involving common pathways such as impaired GABAergic signaling, the atypical development of brainstem arousal systems, or dysfunctional sensory integration networks that influence both movement control and sleep regulation.

Gastrointestinal dysbiosis has been implicated in altered behavior, sleep, and sensory processing. Although this study did not directly measure microbiome composition or metabolomic markers, the observed associations prompt further research into the potential role of intestinal microbiota in modulating motor behaviors such as TW [46,47,48].

### Limitations

This study had some limitations. First, the cross-sectional design does not allow for causal inference. Second, assessments relied in part on parent-reported questionnaires, which may be subject to recall or reporting bias. Third, the sample, although sizable, may not be representative of all children with ASD, particularly those with co-occurring neurological or genetic conditions. Future longitudinal studies and objective gait or autonomic measurements would be beneficial to confirm and extend these findings.

## 5. Conclusions

Taken together, our findings suggest that TW in ASD is not an isolated motor anomaly but part of a complex, multisystemic profile. It appears to be closely intertwined with neurodevelopmental severity, disordered sleep architecture, atypical feeding behaviors, and gastrointestinal dysfunction. The underlying mechanisms may involve a combination of altered central sensory processing, neuroimmune interactions, and microbiota-mediated metabolic signaling. From a clinical perspective, TW should prompt clinicians to assess comorbid sleep and feeding issues and consider an interdisciplinary approach to management. In future research, longitudinal designs and neurophysiological assessments (e.g., EEG graph analysis) may help elucidate the causal relationships and neurobiological substrates underlying these associations.

## Figures and Tables

**Figure 1 medicina-61-01346-f001:**
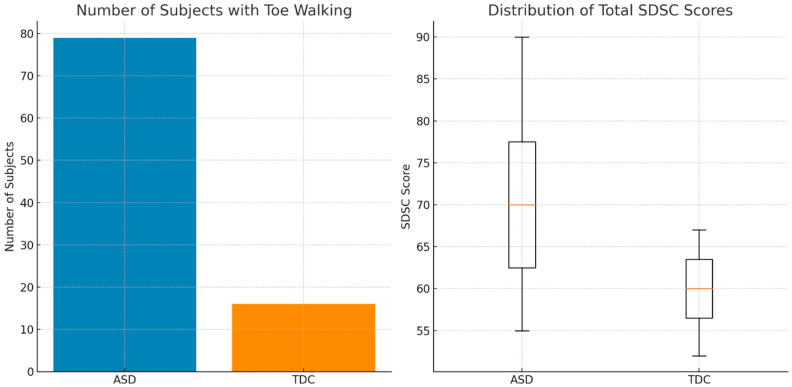
(**Left**): Number of children with toe walking (TW) in the ASD and TDC groups. TW was significantly more common in the ASD group (27.3%) than in controls (5.5%). (**Right**): Boxplot of total SDSC scores, indicating more severe sleep disturbances in the ASD group compared to typically developing controls.

**Figure 2 medicina-61-01346-f002:**
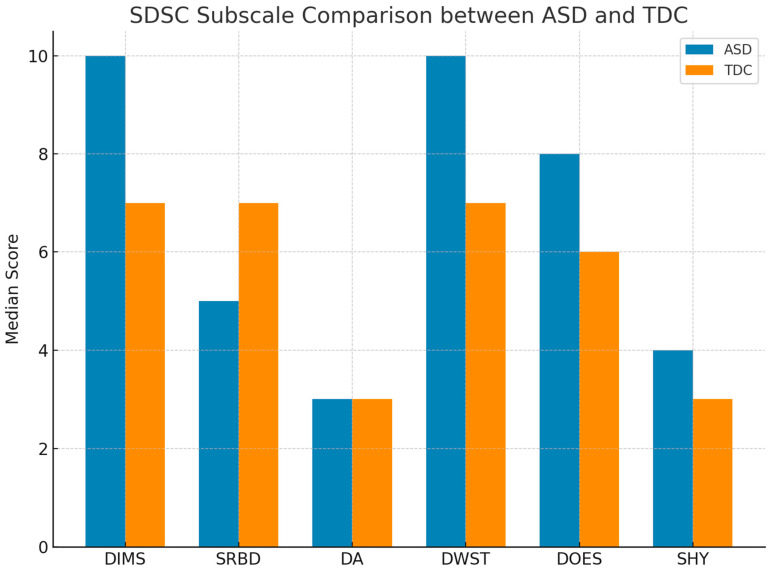
Comparison of median scores for each subscale of the Sleep Disturbance Scale for Children (SDSC) between children with Autism Spectrum Disorder (ASD) and typically developing controls (TDC). Children with ASD showed significantly higher scores in most subscales, particularly in DIMS (Disorders of Initiating and Maintaining Sleep), DWST (Disorders of Wakefulness and Sleep–Wake Transition), and DOES (Disorders of Excessive Somnolence), indicating greater sleep impairment.

**Figure 3 medicina-61-01346-f003:**
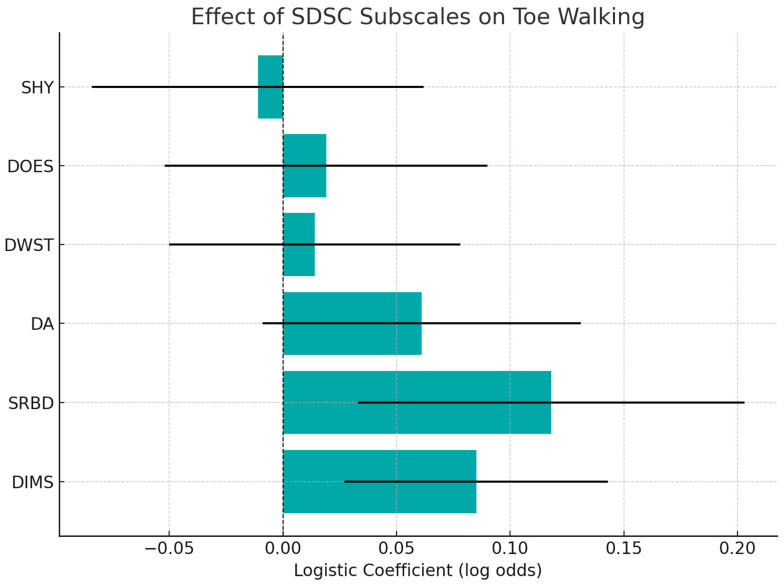
Multivariate logistic regression analysis showing the effect of each SDSC subscale on the likelihood of toe walking (TW) among children with ASD. Although none of the subscales reached statistical significance, DIMS and SRBD (Sleep-Related Breathing Disorders) demonstrated positive trends, suggesting a potential link between sleep disturbance and motor stereotypies.

**Figure 4 medicina-61-01346-f004:**
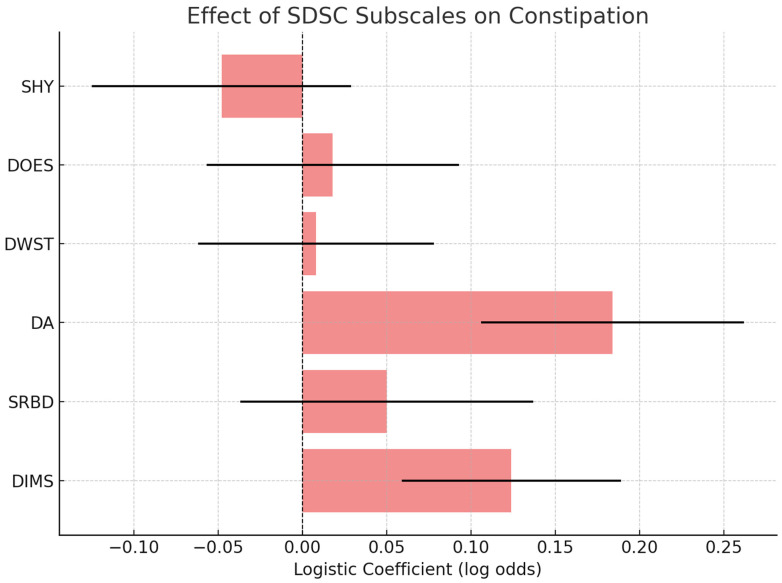
Multivariate logistic regression analysis depicting the influence of SDSC subscales on the presence of constipation in children with ASD. The DA (Disorders of Arousal) subscale emerged as a significant predictor, indicating a possible connection between parasomnias and gastrointestinal dysregulation in more severe ASD profiles.

**Table 1 medicina-61-01346-t001:** Descriptive statistics of the ASD and TDC groups. The table reports the number of participants, mean age (±standard deviation), sex distribution, and the breakdown of ASD severity levels according to DSM-5 classification. Severity level data are not applicable to the TDC group.

Group	N. Subjects	Mean Age (±SD)	Males (%)	Females (%)	ASD Level 1 (*n*/%)	ASD Level 2 (*n*/%)	ASD Level 3 (*n*/%)
**ASD**	289	5.91 ± 0.76	74.4%	25.6%	95 (32.9%)	95 (32.9%)	99 (34.3%)
**TDC**	289	5.93 ± 0.80	74.0%	26.0%	—	—	—

## Data Availability

The original contributions presented in this study are included in the article. Further inquiries can be directed to the corresponding author.

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
