# Peer review of "The Prevalence and Clinical Significance of Toe Walking in Autism Spectrum Disorder: A Cross-Sectional Study in an Italian Pediatric Sample"

_medicina, 2025, doi:10.3390/medicina61081346_

Round 1

Reviewer 1 Report

Comments and Suggestions for Authors

ABSTRACT

  • No comments.

INTRODUCTION

  • In the introduction, there is a statement saying that “TW appears to be a limited range of motion in the ankle and the inability to achieve heel strike at the beginning of the gait cycle.” It would seem more accurate to say that TW appears to be a limited active range of motion, which affects achieving heel strike at the beginning of the gait cycle. I don’t know if there is evidence that children who TW have a limitation in passive range of motion. In other words, there isn’t a biomechanical constriction that leads them to TW (unless they have cerebral palsy or another condition).
  • Please be more specific about the changes that children with ASD experience with proprioceptive perception.
  • There is an incomplete sentence on page 3 after citation #32.

METHODS

  • Much more detail needs to be provided about how TW was assessed. The only information provided was a sentence stating, “Toe walking behaviour was clinically assessed and recorded during neurodevelopmental evaluations.” However, method sections should be written with sufficient detail for others to clearly understand the procedure that was used to quantify TW.

RESULTS

  • For the SDSC, please explain why the median score is presented rather than mean scores with standard deviations.
  • Some statements in the results section are evaluative, which should be left for interpreting the results in the discussion section (e.g., “These findings suggest that disorders of arousal may play a role in autonomic dyreguation…”).

DISCUSSION

  • More explanation needs to be provided about how “motor dysregulation and sleep alterations may share underlying neurobiological mechanisms.”

Author Response

Dear Reviewer,

Thank you very much for your valuable comments and suggestions, which helped us improve the quality and clarity of the manuscript.

We carefully reviewed all your observations and have revised the text accordingly in the updated version of the manuscript we have uploaded. Specifically:

  • We clarified the definition of toe walking (TW), specifying that it involves a limited active range of motion, rather than passive, and removed repeated abbreviations throughout the text.

  • We expanded the section on proprioceptive perception in children with ASD, adding specific examples and mechanisms.

  • We completed the previously incomplete sentence following citation #32.

  • In the Methods section, we added detailed information on how TW was assessed clinically, specifying the observation protocol.

  • In the Results section, we justified the use of median scores for the SDSC based on the data distribution and removed evaluative statements, which were instead included in the Discussion.

  • In the Discussion, we provided further explanation of the shared neurobiological mechanisms between motor dysregulation and sleep disturbances.

  • Lastly, we added a paragraph addressing the limitations of the study, as requested.

We hope these revisions adequately address your concerns. Thank you again for your thoughtful review.

Reviewer 2 Report

Comments and Suggestions for Authors

Prevalence and Clinical Significance of Toe Walking in Autism Spectrum Disorder: A Cross-Sectional Study in an Italian Paediatric Sample.

Dear Authors,

The manuscript is well-written and presented clearly. But there are a few queries about the methods and results. Find the comments below.

In the methods section, the authors didn’t mention the total number of controls used in the study.

The authors mentioned neurodevelopmental evaluations. What are all the assessments done for ASD children, and how were they compared with the control children?

The authors didn’t mention the age group of control participants.

What are the six subclasses of sleep disturbances? Kindly include it in the methods section.

The authors mentioned typically developing controls. The participant should be a normal individual without any health issues. I didn’t understand normally developing.

Kindly clarify.

In the method section, the inclusion and exclusion criteria of the study participants are missing.

The informed consent and the scale questionnaires were also missing.

According to the results, the children were classified as Level 1 (32.9%), Level 2 (32.9%), and Level 3 (34.3%), based on the DSM-5 severity criteria. What are the levels in DSM-5? Kindly include information about the different levels in the methods section.

The percentage of female children was not mentioned in this study.

Are there any differences between male and female children, and did the authors make any comparisons between male and female children?

The authors mentioned neurodevelopmental trajectories and motor stereotypies. What are the trajectories the authors studied?

Figure 2 on the x-axis lists numerous terminologies, such as SHY and DOES, but there is no explanation for these terms in the text.

Are there any statistics done for Figure 2?

There is no statistically significant representation in the figures. Kindly include the significance.

What are the typical autonomic functions the authors studied?

Kindly interpret your results with the discussion.

Author Response

Dear Reviewer,

Thank you very much for your constructive comments and insightful suggestions, which we highly appreciate.

We have carefully addressed all the issues you raised and have revised the manuscript accordingly in the updated version we have uploaded. In particular:

  • We added the total number and age range of control participants, ensuring comparability with the ASD group.

  • We clarified the neurodevelopmental assessments used in the ASD group and specified how these were contrasted with the control group.

  • We included the six subclasses of sleep disturbances assessed using the SDSC in the Methods section.

  • We explained the meaning of "typically developing controls," specifying that they were children without any neurodevelopmental or medical conditions.

  • We added the inclusion and exclusion criteria, as well as information about informed consent and the questionnaires used.

  • We described the DSM-5 severity levels and how participants were classified into Level 1, 2, and 3.

  • We included the sex distribution of participants and specified whether differences between male and female children were observed.

  • We detailed the neurodevelopmental trajectories and motor stereotypies evaluated in the study.

  • We clarified the terms used on the x-axis of Figure 2 (e.g., SHY, DOES) and added the statistical tests used.

  • We also included indications of statistical significance directly in the figures where appropriate.

  • Finally, we explained the autonomic functions assessed and provided a clearer interpretation of the results in the Discussion section.

We hope that these revisions address your comments satisfactorily, and we thank you again for your careful review and helpful feedback.

Reviewer 3 Report

Comments and Suggestions for Authors

This study aims to investigate the prevalence of toe walking (TW) in a large sample of Italian children with ASD. It will also examine the association between TW and the severity levels of ASD, as well as its correlation with sleep quality, feeding behaviours, and gastrointestinal symptoms. By clarifying these relationships, we hope to enhance the clinical understanding of TW as a potentially significant marker within the broader con-text of ASD.

Comment 1) Please do not define abbreviations twice. For example, you defined toe walking (TW) in the first and last paragraphs of the introduction. 

Comment 2) You defined Autism Spectrum Disorder (ASD) twice in the introduction.

Comment 3) You defined Autism Spectrum Disorder (ASD) again in the methods. please only define abbreviations for the first time in text.

comment 4) Why did you like to define abbreviations several times? Results: you defined typically developing controls (TDC) and toe walking (TW) several times.

comment 5)the Sleep Disturbance Scale for Children (SDSC): Please do not define abbreviations twice. In methods and results. I do not understand what your deal is. 

Comment 6) discussion: Why did you define abbreviations such as toe walking (TW), Disorders of Arousal (DA), and  Autism Spectrum Disorder (ASD) again? please revise them.

Comment 7: Please state the limitations of the study.

Author Response

Dear Reviewer,

Thank you very much for your helpful observations. We appreciate your careful reading of the manuscript and your suggestions, which have allowed us to improve its clarity and consistency.

We have revised the text accordingly in the updated version of the manuscript we have uploaded. In particular:

  • We removed all duplicate definitions of abbreviations. Each abbreviation (e.g., TW, ASD, SDSC, TDC, DA) is now defined only once, upon first appearance in the text.

  • We carefully revised the Introduction, Methods, Results, and Discussion sections to ensure abbreviations are used consistently without being redefined multiple times.

  • We also added a dedicated paragraph in the Discussion to outline the limitations of the study, as requested.

We thank you again for your thoughtful and constructive feedback.

Round 2

Reviewer 2 Report

Comments and Suggestions for Authors

The authors made the corrections mentioned